https://doi.org/10.1038/s41467-020-15250-8　　**OPEN**

# Groundwater as a major source of dissolved organic matter to Arctic coastal waters

Craig T. Connolly [1]✉, M. Bayani Cardenas [1], Greta A. Burkart[2], Robert G.M. Spencer[3] & James W. McClelland[1]

Groundwater is projected to become an increasing source of freshwater and nutrients to the Arctic Ocean as permafrost thaws, yet few studies have quantified groundwater inputs to Arctic coastal waters under contemporary conditions. New measurements along the Alaska Beaufort Sea coast show that dissolved organic carbon and nitrogen (DOC and DON) concentrations in supra-permafrost groundwater (SPGW) near the land-sea interface are up to two orders of magnitude higher than in rivers. This dissolved organic matter (DOM) is sourced from readily leachable organic matter in surface soils and deeper centuries-to-millennia-old soils that extend into thawing permafrost. SPGW delivers approximately 400–2100 m$^3$ of freshwater, 14–71 kg of DOC, and 1–4 kg of DON to the coastal ocean per km of shoreline per day during late summer. These substantial fluxes are expected to increase as massive stocks of frozen organic matter in permafrost are liberated in a warming Arctic.

[1] The University of Texas at Austin, Austin, TX 78712, USA. [2] U.S. Fish and Wildlife Service, Arctic National Wildlife Refuge, Fairbanks, AK 99701, USA. [3] Florida State University, Tallahassee, FL 32306, USA. ✉email: craig.connolly@utexas.edu

Groundwater is the largest active reservoir in the global hydrologic cycle and its movement from land to sea represents a major source of freshwater and nutrients for coastal ecological and biogeochemical processes[1]. However, there is little information on direct groundwater nutrient inputs to the coastal ocean in the Arctic[2]. This is partly because of a perception that in northern high-latitude coastal regions permafrost constrains water to flow paths on the land surface. Supra-permafrost groundwater (SPGW) does, however, flow through seasonally thawed active layer soils during the summer and early fall[3,4]. Therefore SPGW has the potential to deliver appreciable quantities of terrestrially-derived nutrients to Arctic coastal waters. SPGW is the principal form of terrestrial groundwater entering nearshore coastal waters in the Arctic since sub-permafrost groundwater flow is firmly separated from the surface by several hundred meters of permafrost[5,6].

In the Arctic, SPGW flow and nutrient transport from soils are tightly coupled because soil water interactions are largely confined to the shallow (typically <1 m), but laterally extensive and highly permeable active layer[7,8]. Soils in northern high-latitude permafrost landscapes contain large amounts of organic matter with a high capacity to release dissolved organic matter (DOM) to aquatic systems[9]. Terrestrial DOM production and export is highest during the spring (May to June) when the thawed portion of the active layer is shallow and snowmelt-driven water flow is confined to near surface organic-rich soils and overlying plant litter layers[10–12]. DOM production and export is lower during the summer (July to October) when active layer thaw exposes deeper soil horizons and groundwater recharged from rainfall and melting ground ice saturates higher proportions of mineral soils[13,14]. Recent studies reveal that SPGW processes govern the summer transfer of DOM to streams and therefore influence riverine export to the coastal ocean[8,15–17].

In contrast to studies conducted on land, very few studies have focused on the role that SPGW plays in the direct transfer of DOM from land to the coastal ocean in the Arctic[2]. Estimates of groundwater inputs to the coastal ocean are needed to support a more complete understanding of what fuels biological production and biogeochemical cycling in Arctic coastal waters. Climate change adds some urgency to this since warming is increasing groundwater discharge across circumpolar regions[18–20], enhancing organic matter decomposition in the active layer, and liberating globally significant stores of organic matter held in high-latitude northern soils and permafrost[21–25]. A baseline understanding of how groundwater mobilizes organic matter held in coastal soils and permafrost is necessary for predicting responses to and feedbacks with climate change.

This study examines the leaching potential of DOM, which includes dissolved organic carbon and nitrogen (DOC and DON), from nearshore Arctic soils and quantifies inputs of SPGW DOM to coastal waters of the eastern Alaska Beaufort Sea (Fig. 1). We first determined the relationship between soil organic carbon (SOC) and nitrogen (SON) contents in different active layer and permafrost soil horizons and the production and leaching potential of DOM from this soil organic matter (SOM). We then determined the relationship between leachable DOM sources and direct SPGW DOM inputs using radiocarbon (14C) dating. Lastly, we estimated DOM fluxes using concentrations of SPGW DOC and DON paired with groundwater discharge estimates that were derived from a steady-state excess radon (222Rn) mass balance model[26–28] (Supplementary Note 1). Data from groundwater and nearby river water were used to compare SPGW and riverine inputs to the Alaska Beaufort Sea coast. To the best of our knowledge this is the first study to quantify fluxes and sources of DOM in direct SPGW inputs to Arctic coastal waters.

## Results and Discussion

**Soil leaching and sources of groundwater DOM.** We found that active layer and shallow permafrost soils along the eastern Alaska Beaufort Sea coast contain 5–20% OC, 0.25–1.3% ON, and produce large quantities of readily leachable DOM. Highest % OC and % ON values were observed within surface soils and plant litter (Table 1). The amount of SOM decreased directly below the organic layer, but increased again in deeper active layer soils and in thawed permafrost samples (Table 1). The range of soil OM contents found herein is typical of tundra active layer and upper permafrost soils[29]. We found a similar pattern in the release of DOM from these soil layers, where surface soils and thawed permafrost tend to have the highest yield (mg DOC and DON per gram of soil) and OC and ON normalized leaching potential (mg DOC and DON per gram of soil OC and ON) (Table 1). These results demonstrate that only a small amount of coastal soil is needed to rapidly produce high concentrations of DOC and DON.

These observations are consistent with the notion that soils in northern high-latitude permafrost landscapes contain large amounts of leachable organic matter that can be exported to aquatic systems. Previous studies have demonstrated that DOM export is highest from plant litter layers and organic-rich soils

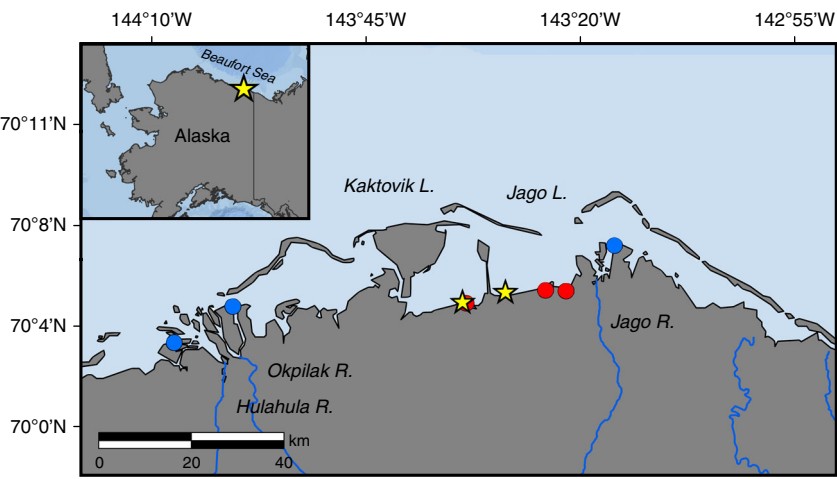

**Fig. 1 Map showing general locations of study sites along the eastern Alaska Beaufort Sea coast.** Samples were collected from soils (red circles), groundwater (yellow stars), and river water (blue circles).

**Table 1 Organic matter content residing in coastal soils and associated soil DOC and DON concentrations from soil water leaching experiments.**

| Sample depth | Soil OM content | | Leachable soil DOM | | | |
|---|---|---|---|---|---|---|
| | % OC (100 × mg C mg soil⁻¹) | % ON (100 × mg N mg soil⁻¹) | mg DOC g soil⁻¹ | mg DON g soil⁻¹ | mg DOC g soil C⁻¹ | mg DON g soil N⁻¹ |
| 0–5 cm | 21.3 ± 4.1 | 1.3 ± 0.24 | 0.48 ± 0.12 | 0.015 ± 0.009 | 2.36 ± 0.52 | 1.10 ± 0.48 |
| 15–20 cm | 5.1 ± 1.0 | 0.25 ± 0.09 | 0.05 ± 0.01 | 0.001 ± 0.001 | 1.06 ± 0.25 | 1.24 ± 1.13 |
| 30–40 cm | 16.8 ± 8.5 | 0.95 ± 0.45 | 0.13 ± 0.05 | 0.006 ± 0.003 | 1.02 ± 0.35 | 1.34 ± 1.03 |
| Permafrost | 9.2 ± 3.7 | 0.58 ± 0.29 | 0.17 ± 0.54 | 0.009 ± 0.004 | 2.68 ± 1.10 | 2.57 ± 1.77 |

The permafrost category represents soils at approximately 5–10 cm below the ice table. Values are ±1 standard error, $n = 3$ for all samples.

**Table 2 Radiocarbon and stable carbon isotopic compositions and C:N ratios in SOC, leachable soil DOC, and SPGW DOC.**

| Sample | Fraction of modern | Δ¹⁴C (‰) | ¹⁴C age (year BP) | δ¹³C (‰) | C:N (molar ratio) |
|---|---|---|---|---|---|
| SOC | | | | | |
| 0–5 cm | 1.046 ± 0.035 | 38 ± 35 | >Modern | −27.8 ± 0.4 | 16.2 ± 0.94 |
| 15–20 cm | 0.622 ± 0.060 | −383 ± 59 | 3895 ± 799 | −28.9 ± 0.5 | 24.2 ± 5.1 |
| 30–40 cm | 0.53 ± 0.053 | −474 ± 53 | 5177 ± 774 | −28.5 ± 0.1 | 17.4 ± 0.87 |
| Permafrost | 0.515 ± 0.515 | −490 ± 57 | 5383 ± 581 | −28.1 ± 0.3 | 17.1 ± 1.4 |
| Leachate DOC | | | | | |
| 0–5 cm | 1.042 ± 0.003 | 33 ± 2.5 | >Modern | −27.3 ± 0.5 | 51.7 ± 15.8 |
| 15–20 cm | 0.749 ± 0.048 | −257 ± 47 | 2350 ± 526 | −27.3 ± 0.1 | 101.5 ± 40.7 |
| 30–40 cm | 0.652 ± 0.023 | −353 ± 23 | 3447 ± 279 | −27.2 ± 0.3 | 44.4 ± 20.0 |
| Permafrost | 0.631 ± 0.022 | −374 ± 21 | 3710 ± 274 | −26.8 ± 0.2 | 26.4 ± 4.51 |
| SPGW DOC | 0.853 ± 0.046 | −154 ± 45 | 1298 ± 421 | −28.2 ± 0.2 | 20 ± 1.2 |

Values are ±1 standard error, $n = 3$ for all samples.

near the surface (<5 cm) because SOM from these horizons have experienced less decomposition and fewer leaching events[30,31]. Lower DOM export is expected from deeper mineral horizons (>20 cm) because these soils have undergone more leaching events, greater cumulative effects of microbial mineralization and passing time, and are more stable because of mineral particle adsorption of DOM[32–35]. However, we found an increase in organic matter content farther down the soil profile, suggesting that cryoturbation (i.e., mixing of the active layer and previously thawed permafrost), leaching, and/or decomposition processes concentrate organic matter near the permafrost boundary. Our results from frozen permafrost samples are consistent with other studies demonstrating that permafrost contains considerable amounts of readily leachable organic matter and releases high concentrations of DOC and DON upon thaw[36–39]. This is in part because permafrost can contain leachates and soils that are only partially decomposed; these soils have been well-preserved and frozen for millennia before thawing[39]. Overall, these trends are markedly similar to patterns in SOC and leachable soil DOC from coastal soils of Elson Lagoon near Utqiagvik (formerly known as Barrow) on the western side of the Alaska Beaufort Sea[40]. Convergence of our findings indicates that soils along the entire Alaska Beaufort Sea coast produce large amounts of leachable DOM that can be exported to lagoons during late summer.

Our analysis of ¹⁴C-DOC (radiocarbon dating of DOC) data reveals that SPGW DOC entering lagoons along the eastern Alaska Beaufort Sea coast is sourced from a combination of surface soils that contain freshly produced organic carbon and deeper soil horizons that may extend into thawing permafrost (Table 2). ¹⁴C-SOC age increased from modern (i.e., the time

period between present day and 1950) near the land surface to ~5400 years before present (yBP; i.e., before 1950) below the permafrost boundary (Table 2). Likewise, leachable soil ¹⁴C-DOC age increased from modern to ~3700 yBP with depth in the shallow permafrost. Interestingly, there is a consistent offset between the ¹⁴C ages of bulk SOC and leached DOC below the surface (>5 cm), indicating that the fraction of readily leachable SOC is younger than the fraction of stable SOC in these deeper soil horizons with stronger mineral particle interactions. We found that groundwater has a ¹⁴C-DOC age of ~1300 yBP (Table 2). In comparison to our ¹⁴C-SOC and soil leachate ¹⁴C-DOC data, it is clear that SPGW DOC must be derived from a combination of organic-rich surface soils and deeper soil horizons.

Taking the analysis a step further, if it is assumed that SPGW receives DOC inputs from each soil horizon in proportion to their leachability (i.e., mg DOC g soil⁻¹; Table 1) and scale these proportions by their observed horizon thickness, we can calculate an expected ¹⁴C-DOC age for SPGW DOC to compare with our measured SPGW DOC age. Here we first calculated proportions of soil-DOC contributions to SPGW DOC (Fig. 2) by multiplying the soil-DOC yields from each of our four soil sections by their observed horizon thickness and divided by the sum of all adjusted soil-DOC yields. The contributing thickness from shallow permafrost was assumed to be 5 cm. We then multiplied each proportion by their respective Δ¹⁴C-DOC values and summed them to estimate what the ¹⁴C-DOC age of the SPGW would be. These calculations were done separately for each core and then averaged (±standard error). The predicted Δ¹⁴C-DOC value of active layer-derived SPGW is −129 ± 30‰ or ~1040 yBP, which is

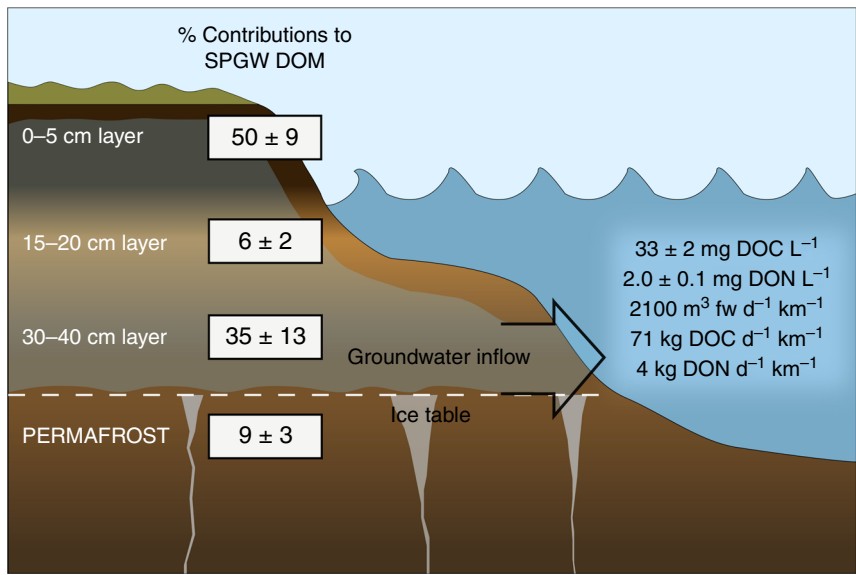

**Fig. 2 Schematic of SPGW flow to Arctic coastal lagoons during late summer summarizing several key results of this study.** (left) Estimates of the percent contributions of soil-DOM from active layer and thawing permafrost soils to SPGW DOM. Percent soil-DOM contributions were made by multiplying soil-DOC yield values of the four indicated soil sections by their horizon thickness and then divided by the sum of the adjusted soil-DOC yields. (right) Average concentrations of DOC and DON found in SPGW and maximum freshwater, DOC, and DON flux values calculated from $^{222}$Rn measurements. Values in the schematic are ±1 standard error.

younger (but within error) of the observed $^{14}$C-DOC age of SPGW. If shallow permafrost contributions are added in, then the expected $\Delta^{14}$C-DOC value is −153 ± 21‰ or ~1270 yBP, which is nearly identical to the observed $^{14}$C-DOC age of SPGW. Given that the $^{14}$C-SOC ages of the 30–40 cm and permafrost sections are markedly similar, these collective deeper soils represent a transient layer that likely experience a range of thaw conditions (from annual to less frequent) near the permafrost-active layer boundary[41]. Although the assumptions in our calculations are simplified, this exercise reinforces the idea that a significant fraction of SPGW DOC is derived from deeper soil layers, including substantial contributions from the transient layer.

It is important to note that SPGW flow through high-centered polygons travels approximately 1 to 10 cm day$^{-1}$ under base flow conditions[42], with shorter expected residence travel times bracketing rain events and along steeper hydraulic gradients near the coastal bluff. Thus, SPGW likely travels from its source to lagoons in timeframes of months to years. Given these travel times, it follows that the groundwater $^{14}$C-DOC ages measured in this study reflect the $^{14}$C age of the soil source of the leached DOC rather than the time since the leached soil DOC formed. That said, we found lower $\delta^{13}$C values and higher C:N ratios in soil leachate DOM compared to SPGW DOM (Table 2). These differences may arise from microbial processing that alters DOM composition during groundwater transport through the active layer[17].

**Concentrations and fluxes of groundwater DOM.** Measurements of DOC and DON demonstrate that SPGW supplies highly concentrated inputs of DOM to Arctic coastal waters during late summer (Fig. 2). These concentrations are one to two orders of magnitude higher than those of nearby rivers during the same timeframe (1.35 ± 0.25 mg C L$^{-1}$; 0.10 ± 0.03 mg N L$^{-1}$; data from ref. [43]). Our SPGW DOC concentration is over an order of magnitude higher than a recent estimate of the global average DOC concentration (2.7 mg C L$^{-1}$) in groundwater aquifers from 15 countries and 4 continents at lower latitudes[44].

Results from a $^{222}$Rn mass-balance model demonstrate that total groundwater discharge to Arctic lagoons is significant during the late summer (Fig. 2). A caveat of this approach is that submarine groundwater discharge estimates do not differentiate terrestrial and marine inputs (i.e., porewater circulating through and discharged from benthic sediments). We found that $^{222}$Rn concentrations in Kaktovik Lagoon were 32.4 ± 3.9 Bq m$^{-3}$ (Supplementary Fig. 1), and that the average SPGW $^{222}$Rn concentration is 223 ± 20 Bq m$^{-3}$. Total groundwater inputs (from land and benthic circulation) to Kaktovik Lagoon is an estimated 8.6 × 10$^5$ m$^3$ day$^{-1}$ or 42.6 m$^3$ day$^{-1}$ m$^{-1}$. These values are similar in magnitude to previously reported values of 18 ± 10 Bq m$^{-3}$ $^{222}$Rn concentration and discharge of 12 ± 4 m$^3$ day$^{-1}$ m$^{-1}$ in Elson Lagoon[2,28].

While the role of groundwater in the coastal ocean has gained marked attention over the past two decades, it has proven challenging to determine how much of this groundwater is terrestrial versus marine derived[1]. Terrestrial-derived groundwater is typically a minor component of total groundwater contributions in the coastal ocean (1–10%), but much higher percentages (e.g., 20–35%) have been observed in coastal systems with strong topographic gradients at the land-sea interface and/or low wave and tidal-driven marine groundwater recirculation[28,45–50]. We expect that the percentage of terrestrial SPGW discharge in our lagoon system is in the lower range of previous estimates (<10%) because Kaktovik Lagoon is surrounded by flat tundra terrain in a region of continuous permafrost. However, terrestrial SPGW discharge is likely greater than 1% because tidal amplitudes and wave activity are very small. Therefore, assuming that 1 to 5% of the groundwater discharge that we measured with $^{222}$Rn in Kaktovik Lagoon is terrestrially-derived, we estimate that SPGW delivers roughly 8.6 × 10$^3$ to 4.3 × 10$^4$ m$^3$ of freshwater day$^{-1}$, 284 kg to 1420 kg of DOC day$^{-1}$, and 17 to 86 kg of DON day$^{-1}$ to Kaktovik Lagoon in late summer. This equates to approximately 400–210 m$^3$ freshwater, 14–71 kg DOC, and 1–4 kg DON per km of shoreline per day in late summer (Fig. 2). Minimum and maximum values associated with these flux estimates range from

59% lower to 76% higher than average estimates (Supplementary Note 2). While quantifying organic matter inputs associated with marine groundwater is beyond the scope of this study, lagoon benthic substrate is largely made up of terrestrial material from runoff and coastal erosion[51], especially in lagoons that contain alluvium that extends to the beaches. Thus, marine-circulated groundwater inputs to coastal lagoons likely include a large amount of terrestrially sourced DOM as well.

Results from the eastern Alaska Beaufort Sea coast provide strong evidence that SPGW can be a substantial source of DOC and DON to Arctic coastal waters. If we assume that SPGW fluxes are similar along the remainder of the Alaska Beaufort Sea coast, we estimate that direct SPGW fluxes range from $0.8–4.2 \times 10^6$ m$^3$ freshwater day$^{-1}$, $28–138 \times 10^3$ kg DOC day$^{-1}$, and $1.7–8.4 \times 10^3$ kg of DON day$^{-1}$ along the entire Alaska Beaufort Sea coastline. For comparison, average August discharge from all rivers draining the North Slope of Alaska is an estimated $8.0 \times 10^7$ m$^3$ day$^{-1}$ (data from 1990–2010 in ref. [52]). Although collective SPGW discharge accounts for only a small percentage of total riverine discharge from the North Slope in August (~1–5%), proportional contributions from SPGW DOM are potentially much more substantial (~14–70% for DOC and ~15–72% for DON) because of the large difference in ground-water versus river water DOC and DON concentrations during late summer. While the relative contributions of direct SPGW versus river inputs are probably minor in areas where major rivers, such as the Colville, flow into the Alaska Beaufort Sea, SPGW inputs may be the dominant source of freshwater, DOC, and DON along extensive stretches of coastline without sizable rivers.

In conclusion, contemporary SPGW fluxes supply substantial quantities of DOC and DON to coastal waters of the Alaska Beaufort Sea and thus represent an underappreciated source of energy for coastal ecosystems in the Arctic. The relative importance of SPGW DOM may be the greatest during late summer when river flow depreciates and groundwater discharge appreciates with maximum thawing of the active layer. The DOM found in SPGW during late summer is sourced from organic matter spanning the entire thawed soil profile, but primarily from highly leachable SOM near the surface and from deeper transient horizons. As warming continues in the Arctic, accelerated permafrost thaw and associated groundwater discharge have the potential to mobilize increasing amounts of soil organic matter.

## Methods

**Water sampling**. SPGW DOM samples were collected along the landward sides of Kaktovik Lagoon on 16–17 August 2014 ($n = 10$) and 8–12 August 2015 ($n = 10$), and then along Jago Lagoon on 17 August 2017 ($n = 15$) (yellow stars in Fig. 1). In total, 35 groundwater DOM samples were collected during the study period (Supplementary Data 1). SPGW was extracted using piezometer wells that were installed to the depth of frozen ground (~1 m) running parallel to the shoreline. Samples from these piezometers capture groundwater that has moved through the active layer to the lagoons without becoming channelized surface water flow. Some samples were also collected along a transect running from the beach to ~50 m inland, as well as from groundwater springs that emerge on the beach as small surface water streams in order to capture SPGW in transit to the coast. Groundwater was collected using a peristaltic-pump system attached with acid washed and Milli-Q rinsed Master-Flex tubing. Groundwater was pumped until clear water was flowing and then filtered through a 0.45 μm GeoTech membrane capsule directly into acid washed and Milli-Q rinsed high-density polyethylene (HDPE) or polycarbonate bottles. Sample bottles were transported in a cooler to the U.S. Fish and Wildlife Arctic National Wildlife Refuge (ANWR) facilities in Kaktovik, Alaska where they were stored frozen until analysis. River water DOM samples were collected previously in August 2011 and 2012 with a similar sampling procedure. SPGW and lagoon water $^{222}$Rn samples were collected using widely used procedures[26–28]. Four discrete groundwater $^{222}$Rn samples were collected along a transect from the beach to the tundra surface at the Kaktovik Lagoon site on 12 August 2015 (Supplementary Data 1). These included three SPGW samples from

piezometers and one from a small groundwater-fed stream. Lagoon water $^{222}$Rn samples were collected from the interior and perimeter areas of Kaktovik Lagoon using a submersible pump deployed near the lagoon bottom on 21 and 22 August 2017 (Supplementary Data 1). The water was directly pumped into a degassing chamber connected to three radon-in-air gas analyzers (Durridge RAD7 connected to the RAD-AQUA module) which analyzed the samples in sequence every 10 min over a 30 min cycle, i.e., each RAD7 measured $^{222}$Rn every 30 min. This work was conducted over two consecutive days in order to sample the entire lagoon.

**Soil and lagoon sediment sampling**. Three soil cores that include the seasonally thawed active layer and shallow permafrost were collected within high-centered polygons (the primary surface type) on the landward sides of Kaktovik Lagoon on 9 August 2015 and Jago Lagoon on 8–9 August 2016 for bulk SOM measurements (red circles in Fig. 1). Three distinct soil layers were evident at all three sites. These included a layer with relatively high organic matter content near the surface (~0–10 cm), a mineral soil layer from ~10–20 cm, and a layer with variable organic matter content from 20 cm to the top of the ice table. Soil samples were collected at four depths: surface–5 cm, 15–20 cm, 30–40 cm, and at 5–10 cm below the frozen boundary. Thaw depths (the depth of the ice table) at the soil collection sites were 40 cm, 53 cm, and 54 cm, meaning permafrost was sampled at respective depths of approximately 45–50 cm, 58–63 cm, and 59–64 cm. These soil increments were chosen specifically in order to gain a wide range of soil types and depths within the active layer and shallow permafrost where changes in organic matter quantity and age are anticipated. The first two samples soil sections (surface–5 cm and 15–20 cm) had an observed horizon thickness of 10 cm, while the next sample soil section (30–40 cm) had observed horizon thicknesses of 20, 33, and 34 cm. Bulk soils were placed into individual whirl-packs, stored frozen at the ANWR facility, and taken back to the University of Texas at Austin, Marine Science Institute (UTMSI) for soil water leaching experiments and chemical analysis. Seven benthic sediment samples were collected from the interior and perimeter areas of Kaktovik Lagoon on 23 August 2017 for $^{222}$Rn activity measurements. Bulk sediments were placed in plastic Ziploc bags, dried, and stored at the University of Texas at Austin until $^{222}$Rn analysis.

**Soil water leaching experiments**. Soil samples (>300 g frozen) were allowed to thaw in a refrigerator (4 °C) until the soils appeared moist, but not dripping with water (at most 24 h). Soils of the same type were then placed on combusted aluminum foil (550 °C for 1 h) and gently homogenized. Care was taken to maintain common soil features and to ensure the sample remained close to its natural condition. This also included removing large roots and anomalous material not representative of the soil horizon. Three subsamples were collected to obtain an average wet weight: dry weight ratio. Subsamples were dried in an oven at 60 °C for 24 h. Thawed soils were kept in the refrigerator for a total of 48 h before leaching. Sample wet weight: dry weight ratios were used to calculate the equivalent of 35 g of dry soil. Field-moist soils were then placed in a combusted glass beaker with 500 mL of a 0.001 N NaHCO$_3$ Nanopure solution (>18.0 Ω-cm), which was used to buffer changes in pH and mimic the natural iconic strength of water in natural systems[30]. The wet soil: solution ratio of the leaching experiments ranged from ~1:10 to 1:4. The yields of DOC per g soil were similar between these soil: solution ratios in a study that took a more rapid leaching approach[40]. We expect no bias in our results related to differences in soil: solution ratios since our leaching experiments were conducted for a longer period of time. The beakers were covered with combusted aluminum foil and stored in a refrigerator at 4 °C. Beakers were shaken intermittently during a 24 h incubation period to simulate groundwater flow through the soil profile. Following, the soil water was filtered through a combusted glass filtration system holding a 0.7 μm glass fiber filter (precombusted at 450 °C >5 h). The filtrate was then dispensed into acid washed and Milli-Q rinsed polycarbonate bottles and stored frozen until DOC and DON concentration and $^{14}$C-DOC (radiocarbon of DOC) analysis. Bulk soils were stored in a drying oven (60 °C) for several weeks before they were finely ground using a mortar and pestle. Ground soil samples went through a vapor fumigation acid/base treatment step to remove inorganic carbon. This step involved storing soil samples in a vacuum-sealed desiccator in a drying oven (60 °C) with a beaker of concentrated HCl for 24 h. Afterwards, soil samples were removed and placed in another vacuum-sealed desiccator with a dish of NaOH pellets, and again stored in a drying oven at 60 °C for another 24 h. This latter step was conducted to neutralize any excess HCl that was not absorbed by the sample.

**Chemical and tracer data collection and analysis**. Concentrations of DOC and total dissolved nitrogen (TDN) were determined using high-temperature oxidation performed on a Shimadzu TOC analyzer fitted with a total nitrogen module for chemiluminescence detection of nitrogen. Inorganic nitrogen (NO$_3^-$ and NH$_4^+$) was analyzed on a Seal-QuAAtro inorganic nutrient analyzer. Concentrations of DON were calculated as the difference between TDN and inorganic nitrogen. NO$_3^-$ measurements made on the Seal-QuAAtro are the equivalent to NO$_3^-$ plus NO$_2^-$. All 35 groundwater samples were measured for DOC, whereas the 20 samples collected in 2014 and 2015 were measured for DON (Supplementary

Data 1). Riverine DOC and DON concentrations near Kaktovik Lagoon were estimated using the average of samples collected in August 2011 from the Hulahula ($n = 1$), Jago ($n = 1$), and Okpilak ($n = 2$) rivers in ref. [43]. In the case of the Okpilak River, two samples were averaged prior. Collective DOC and DON concentrations of north flowing rivers at their outlet from along the North Slope of Alaska were estimated using the average of data from the Turner ($n = 1$), Okpilak ($n = 4$), Hulahula ($n = 1$), Jago ($n = 1$), Canning ($n = 1$), Kuparuk ($n = 5$), Saga-vanirktok ($n = 5$), and Colville ($n = 3$) rivers between July and the end of September 2006–2012 in ref. [43].

We analyzed $^{14}$C-DOC on SPGW samples collected at three individual locations in August 2015 (Supplementary Data 1). DOC samples were prepared for $^{14}$C analysis using the UV-oxidation method at the Woods Hole Oceanographic Institution, National Ocean Sciences Accelerator Mass Spectrometry facility (WHOI/NOSAMS)[53]. Sample water was diluted with pre-treated UV-oxidized nanopure water (to bring the total volume up to 1 L) and placed in a quartz reactor. The combined sample water plus treated nanopure water solution was acidified with 35 g of ultra-high purity (UHP), UV-treated full strength phosphoric acid and then purged with UHP nitrogen gas to remove any inorganic carbon. Pure UHP $O_2$ was subsequently sparged through the system to provide an oxidant for the UV-oxidation of DOC. The sample was then oxidized with UV and the resulting $CO_2$ was transferred to a vacuum line and cryogenically purified. Purified $CO_2$ gas samples were converted to graphite targets by reducing $CO_2$ with an iron catalyst under 1 atm $H_2$ at 550 °C. Bulk soil samples were also high-temperature combusted using an Elementar vario EL Cube C/N analyzer. Bulk soil % OC and % ON were quantified during this step. The resulting $CO_2$ was transferred to a vacuum line and cryogenically purified. The purified $CO_2$ gas samples (soil C) were converted to graphite targets using a closed-tube Zn reduction $CO_2$ graphitization method[54]. Targets were subsequently analyzed for stable and radiocarbon isotopes ($\delta^{13}$C‰ and $^{14}$C as fraction modern carbon). All $\Delta^{14}$C data (in ‰) were corrected for isotopic fraction using measured $\delta^{13}$C values that were quantified during the $^{14}$C-AMS procedure. We measured $\delta^{13}$C in these samples separately on a VG Prism Stable Mass Spectrometer at NOSAMS. $\Delta^{14}$C and radiocarbon age were determined from percent modern carbon using the year of sample analysis according to ref. [55].

We also analyzed $^{14}$C-DOC on a composite SPGW sample that was made from ten individual sample collections on the beach of Jago Lagoon in August 2017. The composite sample collected in 2017 had a $^{14}$C-DOC age of 1060 yBP, which is within error of the mean reported herein. This result was not used along with the individual measurements described in the previous paragraph to calculate an average SPGW age because it was determined using different methodology, but it does confirm that the average $^{14}$C-DOC age estimate used in this study is representative of August SPGW more generally. The groundwater composite collected in August 2017 was solid-phase extracted (SPE-DOM) using modified styrene divinyl benzene polymer PPL cartridges[56]. The SPE-DOM was high-temperature combusted using an Elementar el Vario Cube C/N analyzer and prepared using the same procedure described above. Although SPE-DOM using PPL cartridges typically recovers ~62% of the DOC sample as a salt-free extract[56], we do not anticipate that the SPE-DOM process selectively concentrated compound classes that would result in a different $^{14}$C-DOC value than the UV-oxidation method.

Boat-based $^{222}$Rn measurements were made around Kaktovik Lagoon using a circuit of three RAD7 Radon-in-air monitors (Durridge Co., Inc.) connected to a RAD AQUA device following procedures outlined in ref. [26,27]. The survey started after the system reached full gas/radioactive equilibrium. Measurements at ~1 m depth from the lagoon bottom were made in survey mode while driving a small inflatable boat around the lagoon. To optimize spatial resolution, the three RAD7 units analyzed samples staggered with 15-min intervals with water pumping at flow rates >6 L min$^{-1}$. $^{222}$Rn concentrations shown at each location in Supplementary Figure 1 represent an average of a continuous measurement made along some distance traveled before the average was calculated by each RAD7 instrument. $^{222}$Rn concentrations in discrete SPGW samples were collected using the Wat-250 ml protocol in the Durridge RAD7 user manual. $^{222}$Rn production measurements were conducted with dry lagoon sediments (ranging from 328 to 546 g) using a Durridge RAD7 bulk emissions chamber and methods outlined in the user manual.

**Estimation of groundwater fluxes using $^{222}$Rn.** Naturally occurring $^{222}$Rn ($t_{1/2} = 3.8$ d) is a widely-used geochemical tracer of water flow through rocks and soil/sediments and thus serves a useful tool for quantifying submarine groundwater discharge[1]. Total groundwater discharge was calculated using a steady-state excess $^{222}$Rn mass balance model as described in ref. [26,27]. We used a steady-state model because our lagoon system has no direct connections with the Beaufort Sea and has only two small (<3 m deep) and narrow (~12 m across) passes that connect with adjacent lagoons. It is important to note that there are several primary assumptions of the model: (1) lagoon $^{222}$Rn used for the calculation reflects the lagoon average concentrations over days to weeks; (2) the only significant $^{222}$Rn source is from the tundra active layer and lagoon benthic sediments, and does not include water that enters the lagoon via river inputs; and (3) the only losses of $^{222}$Rn are due to decay and atmospheric evasion. In addition, we assume marine inputs do not affect the $^{222}$Rn inventory in Kaktovik Lagoon since there is very little water exchange with

the Beaufort Sea and lagoon water residence time is long during the summer (weeks to months) in comparison to the decay rate of $^{222}$Rn. The mass-balance model incorporates various end members that include data measured directly by this study, as well as data acquired elsewhere. Minor assumptions regarding this data are outlined in Supplementary Note 1. SPGW $^{222}$Rn concentration was estimated from the average of three collected samples ($223 \pm 20$ Bq m$^{-3}$), while lagoon $^{222}$Rn was estimated from the average of data collected across two surveys ($32.4 \pm 3.85$ Bq m$^{-3}$). $^{222}$Rn production was measured from dry lagoon sediments and then converted to estimates of $^{222}$Rn in pore water ($2932 \pm 650$ Bq m$^{-3}$). Calculations regarding this conversion can be found in Supplementary Note 1. Standard errors in these values reflect variability in concentrations between samples as opposed to analytical uncertainty. A description of the uncertainty in our groundwater discharge estimate can be found in Supplementary Note 2. We did not measure $^{226}$Ra in this study, but rather used estimates from Elson Lagoon in ref. [28]. Therefore we also assume that the $^{226}$Ra concentration between these lagoons is the same. Total groundwater $^{222}$Rn input was estimated using an iterative approach that accounts for the $^{222}$Rn flux from lagoon benthic sediments, atmospheric losses (using wind speed and water temperature data), and the introduction of $^{222}$Rn through the decay of $^{226}$Ra in the lagoon.

## Data availability

New data reported herein is made available in the Supplementary Data 1 file.

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

## Acknowledgements

We thank several people and organizations for their help in this research effort: Ann McNichol, Phil Bennett, Jeff Watson, Ken Dunton, Li Xu, Mary Lardie Gaylord, and Kalina Gospodinova, staff of the Woods Hole Oceanographic Institution (WHOI) NOSAMS facility, the US Fish and Wildlife Service, and the City of Kaktovik. We also thank Natasha Dimova for her helpful feedback. We are very grateful for land access permitted by the Kaktovik Inupiat Corporation. This work was supported by funding from the Geology Foundation at the University of Texas at Austin, a North Pacific Research Board Graduate Research Fellowship, the WHOI NOSAMS Graduate Internship Program, and funding from the National Science Foundation (award 1656026 and OCE-1239667).

## Author contributions

C.T.C., G.A.B., M.B.C., and J.W.M. collected data, contributed to study design, and helped with analysis of groundwater constituents and discharge. C.T.C. collected data and designed the soil sampling, soil water extractions, and isotopic analysis with guidance from R.G.M.S., and J.W.M. C.T.C. prepared the manuscript with contributions from G.A.B., M.B.C., R.G.M.S., and J.W.M.

## Competing interests

The authors declare no competing interests.
