## [Peer Review File · Nature Communications]

Reviewers' comments:

Reviewer #1 (Remarks to the Author):

This is a well thought-out and executed study of the importance of supra-permafrost groundwater discharge and associated DOM flux from the North Slope of Alaska to the Beaufort Sea. The inclusion of soil OM leaching experiments, radiocarbon data, and radon measurements + box model combined raise the bar of this study, providing a comprehensive understanding of the soil horizons and DOM sources (and age) that comprise the groundwater flux. Figure 2 summarizes the study's findings succinctly in an easy to understand way. The paper is well-written.

Minor points of clarification for the authors:

1) How are the error bars provided in Figure 2 for the % contribution to SPGW DOM estimated? More clarification of how they were obtained should be provided in the text for the reader.

2) Please provide a citation(s) for the statement at Line 157-8.

Reviewer #2 (Remarks to the Author):

General Comments

The authors measured DOC and DON concentrations in coastal lagoons, in supra-permafrost groundwater, and in soil core leachates. These measurements were coupled with a ^{14}C measurements of soil and DOC and ^{222}Rn measurements to model the groundwater discharge to the lagoon and fluxes of DOC and DON. Findings suggest that supra-permafrost groundwater may be an important source of freshwater, DOC, and DON to coastal waters. The use of Rn methods in conjunction with C and N cycling is a nice, novel approach. However, I have some concerns about the analyses and calculations in the paper. The authors should provide more text to defend or justify their assumptions and to address uncertainties with their approach. I've made a number of comments below that the authors should consider.

Specific Comments

1. Line 16 – the term “upwards of” implies “more than”, even though the values reported here in the abstract are already the “maximum estimates”

2. Line 78 – The SOM data reported here and in Table 1 are not very informative when reported as %C. Did the authors measure bulk density on soil or sediment cores? It would be more informative to report SOM as C or N densities (gC cm⁻³ or gN cm⁻³). As an FYI, there are many organic-soil types that have higher C contents than these surface soils, often exceeding 35-40%.

3. Line 83 – I would recommend not using the term “production” here, as it’s not a rate measured over time. I recommend changing the text to DOC “yield” or “amount of water soluble DOC”

4. Line 96 – Change wording to simply “mineral horizons”, not “mineral-rich, soil horizons”. By definition, mineral soils have less C (<20% C) than organic soils (>20%), thus it makes sense that they would yield less C.

5. Line 97 – What do you mean by “enhanced microbial mineralization over time”? Deeper mineral soils are typically colder than surface soils, and these cold temperatures are a primary constraint on microbial decomposition of organic matter in soils.

6. Line 104 – Please clarify what you mean by “permafrost can contain processed leachates”?

7. Figure 2 – It’s not clear where the values for “% contributions to SPGW DOM” come from? In the legend it says inputs were “estimate from measurements”, but measurements of what? It’s also not clear from the figure or the legend what the values on the right represent: “estimated upper range of fluxes?” The legend needs much more detail.

8. Lines 123-127 – What rationale could you provide to support the assumption that “SPGW receives DOM inputs from each soil horizons that is proportional to their amount of leachable soil-DOM”? Why not do something more quantitative like a two-isotope (14C, 13C) mixing analysis to determine contemporary and older source contribution to SPGW DOC? See, as an example, Mann et al. (2015) in Nature Communications. Is there any reason to believe the water flux from each horizon is the same in situ as it was controlled for in the leaching experiment? Further, it’s still not clear entirely how the authors came up with the estimate/predicted 14C-DOC value here.

9. Lines 127-133 – I don’t understand how these calculations was made either. The authors need to clarify the text here and justify their approach.

10. Line 135 – Omit “peri-permafrost”. This transient layer has already been defined in the literature. The authors should refer to and cite Shur et al. 2005 in Permafrost and Periglacial Processes (<https://onlinelibrary.wiley.com/doi/abs/10.1002/ppp.518>).

11. Lines 167-168 – How are we to evaluate if groundwater discharge is “significant” or not? How does the magnitude of GW discharge compare to other components of the hydrologic budget of the lagoon? If the method cannot tease apart marine vs SPGW inputs, what confidence can we place on the role of SPGW? How do GW inputs compare to surface water inputs or precip or ET?

12. Line 194-195 – Can you provide a citation for this statement about “lagoon benthic substract is largely made up of terrestrial material...”?

13.Lines 258-260 – The authors should describe soils in a more scientific manner (“mineral-like”? “muck”?) There are defined protocols for describing soils using either USDA or Canadian publications, based on texture, color, inclusions, etc.

14.Line 291 – no need to hyphenate “soil-water”

15.Line 294 – need to define “DO14C” as “radiocarbon of DOC”. From a stylistic perspective, I prefer 14C-DOC, which you actually use in other section of the manuscript.

16.Line 308 – the text here is confusing...simply state DON was calculated as the difference between TDN and inorganic nitrogen.

17.Lines 312-317 – This is a pretty low sample size...not sure how well this sampling represents the “collective riverine DOC and DON concentrations for the entire North Slope of Alaska”?

18.Line 319 – so n=3 14C-DOC samples??

19.Line 341 – What do you mean by “Underway”?

Reviewer #3 (Remarks to the Author):

This is an excellent paper addressing a timely topic of broad international interest. The authors used appropriate methods including a comprehensive uncertainty analysis that is unfortunately uncommon in most other groundwater tracer investigations. The conclusions are sound and well backed by data. The dataset is of high quality and I am impressed on how the authors were able to achieve this in very challenging field conditions. The paper is consistent with earlier observations in Alaska demonstrating significant seepage of soil carbon in the Arctic, but it goes much farther by using radiogenic carbon to estimate the age of carbon in seeping groundwater. This represents a paradigm shift in the field since most colleagues have put groundwater discharge in the too hard basket for years, and have focused on river inputs into the Arctic. I expect this paper will encourage the creation of a new line of research and follow up investigations. The conceptual model is simple but insightful. I would be using this model as a teaching resource. I have very little to criticize and would recommend publication as is or after minor review.

I would have only a few minor comments or suggestions for improvement:

1) The author’s estimate of fresh SGD to be 1-5% is somewhat rough (Line 185). Can you develop a salt balance similar to the radon mass balance to get additional insight into fresh SGD? Maybe obtain some insight from radon versus salinity relationships qualitatively supporting a small contribution of fresh SGD?

2) The radon mass balance model assumes steady state. Can you offer some more hydrogeological insight to justify the assumption?

3) Supplementary Figure 1 shows a radon distribution map. I wonder if the hotspots can be explained by local features. Can you overlay any important local features on this map? Do the radon hotspots match DOC hotspots of freshwater?

Response to Reviewer 1 Comments:

This is a well thought-out and executed study of the importance of supra-permafrost groundwater discharge and associated DOM flux from the North Slope of Alaska to the Beaufort Sea. The inclusion of soil OM leaching experiments, radiocarbon data, and radon measurements + box model combined raise the bar of this study, providing a comprehensive understanding of the soil horizons and DOM sources (and age) that comprise the groundwater flux. Figure 2 summarizes the study's findings succinctly in an easy to understand way. The paper is well-written.

We would like to thank reviewer one for their positive comments and helpful feedback.

Minor points of clarification for the authors:

1) How are the error bars provided in Figure 2 for the % contribution to SPGW DOM estimated? More clarification of how they were obtained should be provided in the text for the reader.

In our revised manuscript we did a much more thorough job explaining how % contributions to SPGW DOM were estimated, including how the standard errors were calculated. The revised text can be found in lines 132-141.

2) Please provide a citation(s) for the statement at Line 157-8.

We added a citation here.

Response to Reviewer 2 Comments:

The authors measured DOC and DON concentrations in coastal lagoons, in supra-permafrost groundwater, and in soil core leachates. These measurements were coupled with a ^{14}C measurements of soil and DOC and ^{222}Rn measurements to model the groundwater discharge to the lagoon and fluxes of DOC and DON. Findings suggest that suprapermafrost groundwater may be an important source of freshwater, DOC, and DON to coastal waters. The use of Rn methods in conjunction with C and N cycling is a nice, novel approach. However, I have some concerns about the analyses and calculations in the paper. The authors should provide more text to defend or justify their assumptions and to address uncertainties with their approach. I've made a number of comments below that the authors should consider.

We would like to thank reviewer two for their helpful considerations and positive feedback. We have addressed their specific concerns below.

Specific Comments

1. Line 16 – the term “upwards of” implies “more than”, even though the values reported here in the abstract are already the “maximum estimates”

This is a good point and we edited the text accordingly in line 29.

2.Line 78 – The SOM data reported here and in Table 1 are not very informative when reported as %C. Did the authors measure bulk density on soil or sediment cores? It would be more informative to report SOM as C or N densities (gC cm⁻³ or gN cm⁻³). As an FYI, there are many organic-soil types that have higher C contents than these surface soils, often exceeding 35-40%.

Unfortunately we did not measure bulk density, otherwise we agree that this would be another useful way to express our SOM data. In any case, % OC is also commonly reported in papers that focus on soils and sediments (e.g., Michaelson et al., 1996; Schuur et al., 2008). Thus, we are confident that reporting our SOM data in % OC will be helpful for other studies to compare to. We recognized that the range of values measured in this study is pretty typical of tundra active layer and upper permafrost soils. We revised the text accordingly in lines 85-90.

3.Line 83 – I would recommend not using the term “production” here, as it’s not a rate measured over time. I recommend changing the text to DOC “yield” or “amount of water soluble DOC”

Good point. In line 92 we replaced “production” with “yield.”

4.Line 96 – Change wording to simply “mineral horizons”, not “mineral-rich, soil horizons”. By definition, mineral soils have less C (<20% C) than organic soils (>20%), thus it makes sense that they would yield less C.

In line 102 we changed the wording to “mineral horizons.”

5.Line 97 – What do you mean by “enhanced microbial mineralization over time”? Deeper mineral soils are typically colder than surface soils, and these cold temperatures are a primary constraint on microbial decomposition of organic matter in soils.

Here our intent was to convey that deeper and older mineral soils are more degraded by microbes. We now realize that in its previous form the sentence suggested that microbial decomposition has increased over time, which is not what we aimed to say. Thank you for bringing this up. To clarify, we revised the text in line 103 to read “greater cumulative effects of microbial mineralization with passing time.”

6.Line 104 – Please clarify what you mean by “permafrost can contain processed leachates”?

Here we intended to convey that microbes have already processed these soils/leachates to some degree. In lines 110-112, we revised the text to read “This is in part because permafrost can contain leachates and soils that are only partially decomposed; these have been well-preserved and frozen for millennia before thawing.”

7.Figure 2 – It’s not clear where the values for “% contributions to SPGW DOM” come from? In the legend it says inputs were “estimate from measurements”, but measurements of what? It’s also not clear from the figure or the legend what the values on the right represent: “estimated upper range of fluxes?” The legend needs much more detail.

Please see the revised figure for more clarity. In addition, please see lines 132 to 141, which include more detail on how our calculations were made to estimate ^{14}C -DOC values of the soil profile and % soil-DOM contributions to SPGW DOM.

8.Lines 123-127 – What rationale could you provide to support the assumption that “SPGW receives DOM inputs from each soil horizons that is proportional to their amount of leachable soil-DOM”? Why not do something more quantitative like a two-isotope (^{14}C , ^{13}C) mixing analysis to determine contemporary and older source contribution to SPGW DOC? See, as an example, Mann et al. (2015) in Nature Communications.

Early in our data analysis and paper development effort we used two-tracer mixing model calculations within a Bayesian framework to estimate proportional contributions of soil-DOM from each soil horizon to overall SPGW DOM. We did this using $\Delta^{14}\text{C}$ and $\delta^{13}\text{C}$ data. However, we ultimately decided against a two-source mixing model approach because our soil-DOM $\delta^{13}\text{C}$ data are very similar for each soil horizon. Our $\delta^{13}\text{C}$ values are not distinct enough to justify using them meaningfully for two-tracer Bayesian mixing model calculations. Those calculations did confirm that contributions from deeper soil layers were required to produce the signature that we measured in groundwater, and that contributions were likely coming from all of the layers that we measured, but the results were overwhelmingly driven by the spread in ^{14}C ages with depth.

As an alternative approach, we used our experimental soil-DOM leaching rates, soil layer thickness data, and associated ^{14}C -DOC ages to estimate the expected ^{14}C -DOC age of the mixture of soil-DOM and compared this result with our measured ^{14}C -DOC age of SPGW. In other words, we asked: can the measured age of SPGW be accounted for by soil layer contributions that are proportional to their different ages, leaching rates, and layer thicknesses. The answer to that question is yes. This does not rule out the possibility of other combinations, but neither does a Bayesian mixing model approach. What our calculations do demonstrate is that the age of the SGPW is not markedly younger or older than what would be expected from soil layer contributions that are proportional to leaching rates.

Is there any reason to believe the water flux from each horizon is the same *in situ* as it was controlled for in the leaching experiment?

Unfortunately our study is limited in its ability to tease apart the water flux from each soil horizon *in situ* and how this compares to our controlled leaching experiment. We agree that there are a number of factors in nature that would ideally be accounted for in controlled leaching experiments (e.g., water-soil contact time and total water-flux as mentioned). Our intention with these calculations was to provide first-cut estimates of contributions from different soil layers to SPGW DOM. Our discussion does not focus on exact proportions, but rather emphasizes the more general conclusion that DOM in SPGW is not simply derived from near-surface soil organic matter stocks. Nonetheless, we feel that the percentage contribution estimates provided in Figure 2 will be instructive to readers and provide a benchmark for comparison in future studies.

Further, it's still not clear entirely how the authors came up with the estimate/predicted 14C-DOC value here.

We provided more detail, which can be found in lines 132-141.

9.Lines 127-133 – I don't understand how these calculations was made either. The authors need to clarify the text here and justify their approach.

Please see our comment above.

10.Line 135 – Omit “peri-permafrost”. This transient layer has already been defined in the literature. The authors should refer to and cite Shur et al. 2005 in Permafrost and Periglacial Processes (<https://onlinelibrary.wiley.com/doi/abs/10.1002/ppp.518>).

We agree that it would be better to reference the Shur et al., 2005 article. We made this adjustment in line 148.

11.Lines 167-168 – How are we to evaluate if groundwater discharge is “significant” or not? How does the magnitude of GW discharge compare to other components of the hydrologic budget of the lagoon? If the method cannot tease apart marine vs SPGW inputs, what confidence can we place on the role of SPGW? How do GW inputs compare to surface water inputs or precip or ET?

In lines 205-214, we do provide a comparison between SPGW and river inputs from the North Slope of Alaska in August. Here we demonstrate that while SPGW discharge is relatively small compared to rivers draining the North Slope of Alaska, SPGW fluxes of DOM are likely much larger due to the high concentrations of SPGW DOC and DON.

Unfortunately there is little hydrologic data (or papers published) on the lagoons specifically to do these other detailed comparisons. While we recognize that there is climate-modeled precipitation and ET data available for the North Slope, we chose not to dive into this level of detail in our paper as to keep the focus on the main story about SPGW as a source of DOM to Arctic coastal waters.

We recognize that measuring Rn does not tease apart SPGW versus marine inputs, although many studies have examined the fraction of land-derived groundwater discharge to the ocean elsewhere in the world. In lines 181-188, we make a conservative estimate of the proportion of total groundwater discharge that is land-derived (i.e., SPGW: 1-5 %). Our estimate was based on a literature review and observations of our lagoon systems. Although a more discrete number would be ideal, we are confident that this range is representative of the possible fraction of SPGW in total groundwater discharge measured by Rn.

12.Line 194-195 – Can you provide a citation for this statement about “lagoon benthic substrate is largely made up of terrestrial material...”?

Yes and we added this reference in line 197.

13.Lines 258-260 – The authors should describe soils in a more scientific manner (“mineral-like”? “muck”?) There are defined protocols for describing soils using either USDA or Canadian publications, based on texture, color, inclusions, etc.

We agree that a soil classification would be ideal here. Since we did not make specific measurements to identify soil class, we revised the text in lines 260-262 and now describe the soils (and their transitions) in a manner that is more relative to our study (and which we have measurements for).

14.Line 291 – no need to hyphenate “soil-water”

We eliminated the hyphen here and elsewhere in the manuscript.

15.Line 294 – need to define “DO14C” as “radiocarbon of DOC”. From a stylistic perspective, I prefer 14C-DOC, which you actually use in other section of the manuscript.

We agree that 14C-DOC is a better way to represent “radiocarbon of DOC”. We revised the text here to reflect this as well as where 14C-DOC is first mentioned in the manuscript.

16.Line 308 – the text here is confusing...simply state DON was calculated as the difference between TDN and inorganic nitrogen.

Thank you for the consideration. We revised the text here as suggested.

17.Lines 312-317 – This is a pretty low sample size...not sure how well this sampling represents the “collective riverine DOC and DON concentrations for the entire North Slope of Alaska”?

We mean to say that these samples reflect the collective DOC and DON concentrations of north flowing rivers at their outlet along the North Slope of Alaska, of which there are far fewer samples compared to streams and rivers farther inland. We revised the text in lines 318-321 to make this clearer.

18.Line 319 – so n=3 14C-DOC samples??

Yes, we sampled three different locations for 14C-DOC measurements in groundwater. Radiocarbon analyses are very expensive, and this sample size is not unusual for a 14C study. As a reality-check, we also measured ¹⁴C-DOC on a composite groundwater sample that was made from 10 individual collections. The age of this composite sample was similar to the average from our three discrete measurements. We did not include this information in the original submission because the composite was collected during a different year and the DOM from this composite was prepared using a different method (solid-phase extraction). However, in response to the question about sample number raised here, we decided to go ahead and include information about our composite analysis. We have added a paragraph about this in lines 345-357.

19.Line 341 – What do you mean by “Underway”?

We deleted the word “Underway” at the start of this sentence.

Response to Reviewer 3 Comments:

This is an excellent paper addressing a timely topic of broad international interest. The authors used appropriate methods including a comprehensive uncertainty analysis that is unfortunately uncommon in most other groundwater tracer investigations. The conclusions are sound and well backed by data. The dataset is of high quality and I am impressed on how the authors were able to achieve this in very challenging field conditions. The paper is consistent with earlier observations in Alaska demonstrating significant seepage of soil carbon in the Arctic, but it goes much farther by using radiogenic carbon to estimate the age of carbon in seeping groundwater. This represents a paradigm shift in the field since most colleagues have put groundwater discharge in the too hard basket for years, and have focused on river inputs into the Arctic. I expect this paper will encourage the creation of a new line of research and follow up investigations. The conceptual model is simple but insightful. I would be using this model as a teaching resource. I have very little to criticize and would recommend publication as is or after minor review.

We would like to thank reviewer three for their very positive comments and helpful feedback.

I would have only a few minor comments or suggestions for improvement:

1) The author’s estimate of fresh SGD to be 1-5% is somewhat rough (Line 185). Can you develop a salt balance similar to the radon mass balance to get additional insight into fresh SGD? Maybe obtain some insight from radon versus salinity relationships qualitatively supporting a small contribution of fresh SGD?

This is an interesting idea. However it would be difficult to conduct a salt balance or examine radon versus salinity relationships given that our lagoon system contains freshwater inputs from sea ice melt and rivers in addition to supra-permafrost groundwater during the summer. In any case, we don’t have a spatially-comprehensive salinity dataset to develop a salt balance with.

2) The radon mass balance model assumes steady state. Can you offer some more hydrogeological insight to justify the assumption?

This is a good question that we could have done a better job at explaining in the text. We decided to use a steady-state mass balance model because Kaktovik Lagoon has no direct connections with the Beaufort Sea and has only two small (< 3m deep) and narrow (~ 12m across) passes that connect to adjacent lagoons. In other words, we can treat Kaktovik Lagoon as a steady-state system (similar to a lake) because it has very little (if any) tidal intrusion and marine influence over the short time period in which Rn decays. In lines 377-379, we explain this better to justify our assumption.

3) Supplementary Figure 1 shows a radon distribution map. I wonder if the hotspots can be explained by local features. Can you overlay any important local features on this map? Do the radon hotspots match DOC hotspots of freshwater?

Yes, we do think that these hotspots might be caused by some effect of local features. For instance, we found higher Rn on the western side of Kaktovik Lagoon, which are surrounded by tall (~ 10 ft) eroding permafrost bluffs with potentially stronger hydraulic gradients and more groundwater inflow than other areas draining into the lagoon. We also found higher Rn in the southwestern corner of the lagoon, which is adjacent to a wetland. This could also be the source of higher groundwater derived Rn to this area. There are maps that might be able to show some of these local regions, however, their resolution is often very coarse at the small scales we observe these features at in this remote region of coastal Alaska. We appreciate the interest in adding more detail to this supplementary figure, so we decided to add some text for our readers to think about further. Unfortunately we do not have DOC data in Kaktovik Lagoon that are co-located with our Rn measurements. This would be another excellent component of a follow up study.

REVIEWERS' COMMENTS:

Reviewer #2 (Remarks to the Author):

The authors have done a great job of responding to reviewer comments. I have no further comments, and believe the manuscript is now suitable for publication.

Reviewer #3 (Remarks to the Author):

I am satisfied with the reviewers response to my comments and recommend the paper is accepted as is.